# Impact of Surface and Subsurface-Intensified Eddies on Sea Surface Temperature and Chlorophyll-a in the North Indian Ocean Utilizing Deep Learning

Yingjie Liu[1], Xiaofeng Li[1]

[1]CAS Key Laboratory of Ocean Circulation and Waves, Institute of Oceanology, Chinese Academy of Sciences, Qingdao 266071, China.

*Correspondence to*: Xiaofeng Li (lixf@qdio.ac.cn)

**Abstract.** Mesoscale eddies, including surface-intensified eddies (SEs) and subsurface-intensified eddies (SSEs), significantly influence phytoplankton distribution in the ocean. Nevertheless, due to the sparse in-situ data, it is still unclear in understanding the characteristics of SSEs and their influence on chlorophyll-a (Chl-a) concentration. Consequently, the study utilized a deep learning model to extract SEs and SSEs in the North Indian Ocean (NIO) from 2000 to 2015, using satellite-derived sea surface height (SSH) and sea surface temperature (SST) data. The analysis revealed that SSEs accounted for 39% of the total eddies in the NIO, and their SST signatures exhibited opposite behavior compared to SEs. Furthermore, by integrating ocean color remote sensing data, the study investigated the contrasting impacts of SEs and SSEs on Chl-a concentration in two basins of the NIO: the Arabian Sea (AS) and the Bay of Bengal (BoB), known for their disparate biological productivity. In the AS, SEs induced Chl-a anomalies that were two to three times higher than those caused by SSEs. Notably, there were no significant differences in Chl-a anomalies induced by the same type of eddies between summer and winter. In contrast, the BoB exhibited distinct seasonal variations, where SEs induced slightly higher Chl-a anomalies than SSEs during the summer, while substantial differences were observed during the winter. Specifically, subsurface-intensified anticyclonic eddies (SSAEs) led to positive Chl-a anomalies, contrasting the negative anomalies induced by surface-intensified anticyclonic eddies (SAEs) with comparable magnitudes. Moreover, while both subsurface-intensified cyclonic eddies (SSCEs) and surface-intensified cyclonic eddies (SCEs) resulted in positive Chl-a anomalies during winter in the BoB, the magnitude of SSCEs was only one-third of that induced by SCEs. Besides, subsurface Chl-a induced by SSAEs (SSCEs) is ~ 0.1 mg/m$^3$ greater (less) than that caused by SAEs (SCEs) in the upper 30 (50) m using Biogeochemical Argo profiles. The distinct Chl-a between SEs and SSEs can be attributed to their contrasting subsurface structures revealed by Argo profiles. Compared to SAEs (SCEs), SSAEs (SSCEs) enhance (decrease) production via the convex (concave) of the isopycnals that occur around the mixed layer. The study provides a valuable approach to investigating subsurface eddies and contributes to a comprehensive understanding of their influence on chlorophyll concentration.

# 1 Introduction

Mesoscale eddies are widely exist in the global ocean (Chen and Han, 2019; Chen et al., 2021; Chelton et al., 2011a; Faghmous et al., 2015), which significantly influence phytoplankton distribution through several processes, including eddy stirring (Chelton et al., 2011b), eddy trapping (Lehahn et al., 2011); eddy upwelling and downwelling (Gaube et al., 2013); eddy-induced Ekman pumping (Gaube et al., 2014; Siegel et al., 2011; Gaube et al., 2013), and eddy strain-induced pumping (Zhang et al., 2019). Previous studies predominantly focused on investigating chlorophyll distribution induced by surface-intensified

eddies (SEs), which can be generally classified into surface-intensified anticyclonic eddies (SAEs) and surface-intensified cyclonic eddies (SCEs) based on their rotation direction (Chen et al., 2019). It is important to note that mesoscale eddies can be further subdivided into distinct categories by the location of their core, where the potential vorticity reaches its maximum. The core can be located in the surface or subsurface layers (Assassi et al., 2016), resulting in SEs or subsurface-intensified eddies (SSEs). SSEs are conjectured to be due to eddy–wind interaction, local adiabatic processes, barotropic and baroclinic

instabilities, or topographic influences (Badin et al., 2011; Meunier et al., 2018; Thomas, 2008; Mcgillicuddy, 2015). Due to the particular lens-like structure of isopycnals, SSEs are an important supplier of nutrients for the euphotic zone and greatly enhance primary production (Mcgillicuddy Jr et al., 2007; Ledwell et al., 2008; Karstensen et al., 2017). SSEs have been observed in various ocean regions using in-situ data, such as the California Undercurrent eddies in the northeastern Pacific (Garfield et al., 1999), the Mediterranean water eddies and slope water oceanic eddies in the northeastern Atlantic

(Bashmachnikov et al., 2013; Paillet et al., 2002). However, the sparse availability of in-situ data makes it challenging to determine whether the eddies observed in satellite-derived maps are subsurface-intensified. Therefore, there is still uncertainty for further research regarding the characteristics of SSEs and their impact on chlorophyll concentration.

The surface and interior ocean are highly correlated, and the subsurface signals in the ocean can be reflected at the surface (Klemas and Yan, 2014). The relationship between sea surface height (SSH) and sea surface temperature (SST) within eddies

has proven to be an effective index for differentiating between SEs and SSEs using multi-source remote sensing data (Assassi et al., 2016; Bashmachnikov et al., 2013; Caballero et al., 2008), which has demonstrated successful application across diverse oceanic regions (Wang et al., 2019; Greaser et al., 2020; Trott et al., 2019). However, SST and SSH within eddies are subject to the intricate influence of multiple physical processes, leading to the intricate and nonlinear SST-SSH relationship that traditional statistical methods may not adequately capture. The deep learning (DL) technique has recently demonstrated

remarkable capabilities in analyzing and extracting intricate patterns and relationships from multi-source big data (Ham et al., 2019; Lecun et al., 2015; Lu et al., 2019; Su et al., 2015; Jiang et al., 2022; Su et al., 2021a), enabling a deeper and more comprehensive exploration of the intricate dynamics within SEs and SSEs.

Mesoscale eddies are prominent features in the North Indian Ocean (NIO) (Zhan et al., 2020; Trott et al., 2019; Chen et al., 2012; Gulakaram et al., 2020; Greaser et al., 2020), which consists of the Bay of Bengal (BoB) and the Arabian Sea (AS), two

distinct basins that exhibit substantial differences in terms of their biological productivity. In the NIO, intense southwesterly summer monsoon winds blow between June and September, while relatively weak northeasterly winter winds blow between

November and February (Prasad, 2004). Besides, the winds over the AS are stronger than the BoB due to the Findlater Jet during the summer monsoon (Findlater, 1969). The intense summer monsoon makes the AS one of the world's most productive regions (Kumar et al., 2002), with various physical processes contributing to its productivity, such as open ocean upwelling

(Brock et al., 1991), wind-driven mixing (Lee et al., 2000), lateral advection (Kumar et al., 2001), and the coastal upwelling along Somalia (Kumar et al., 2002). Conversely, the BoB is regarded as a region with lower biological productivity due to weaker summer monsoon and lower salinity (Kumar et al., 2002; Prasad, 2004). Previous literature mainly investigates the influence of SEs on biological features in the AS and the BoB (Yang et al., 2020; Shafeeque et al., 2021; Smitha et al., 2022). However, both SEs and SSEs were found in the AS (Trott et al., 2019) and the BoB (Greaser et al., 2020; Babu et al., 1991).

For example, during the southwest monsoon seasons from 2015 to 2018, Trott et al. (2019) found that 38.6% of anticyclonic eddies are subsurface-intensified, and 28.5% of cyclonic eddies are subsurface-intensified in the AS. Considering that the number of SSEs cannot be ignored, further investigations are needed to examine the effects of SSEs on chlorophyll distribution in the NIO.

Therefore, the study proposes a DL-based model to distinguish between SEs and SSEs using satellite-derived altimetry SSH

and infrared SST data. Consequently, the study conducts a comparative analysis to assess the differential impacts of SEs and SSEs on chlorophyll concentrations in the NIO. Section 2 introduces the satellite-derived data, in-situ data, and methods to distinguish and analyze SEs and SSEs. Section 3 examines and contrasts the spatial characteristics and seasonal variations of SST and chlorophyll anomalies caused by SEs and SSEs in the AS and the BoB. Section 4 of the study focuses on constructing subsurface eddy structures using in-situ data to validate the DL-based model's accuracy and explain the differences in

chlorophyll distribution caused by SEs and SSEs. In Section 5, the study presents its conclusions based on the findings and analysis conducted throughout the research.

## 2 Data and Methods

### 2.1 Data

#### 2.1.1 Satellite-derived Dataset and Products

The SSH anomalies (SSHA) dataset is obtained from the European Copernicus Marine Environment Monitoring Service (Pujol et al., 2016). The dataset is derived by combining data from multiple altimeter missions and is available daily. The spatial resolution of the dataset is 0.25°, providing detailed information about the variations in SSH across the study region. A spatial filter with half-power filter cutoffs of 20° longitude by 10° latitude is applied to the SSHA map to facilitate the detection of eddies (Chelton et al., 2011a). The SST dataset used in the study is the NOAA Optimum Interpolation (OI) SST product from

Reynolds et al. (2007). The dataset is available daily and has a spatial resolution of 0.25°. To identify eddy‐induced SST anomalies (SSTA), temporal and spatial filters were applied to the SST field. The temporal filter utilized a band-pass Butterworth window to preserve the temporal signal within 7-90 days. The filter is chosen based on the typical lifetimes of

eddies in the NIO, ensuring that the relevant temporal variations associated with eddy dynamics are captured. Meanwhile, the spatial filter employed a moving average Hann window to retain spatial scales smaller than 600 km. These filters have been shown to provide robust results for obtaining mesoscale SSTA field (Bôas et al., 2015).

In addition, the ocean color observed chlorophyll-a (Chl-a) product is used to evaluate chlorophyll concentrations induced by eddies. The daily Chl-a dataset of 4 km was produced by the European Space Agency (Maritorena et al., 2010). The Chl-a measurements were averaged onto the 0.25°grid as the SSHA observations. The unit for Chl-a concentration is mg/m$^3$, and Chl-a values are firstly log-transformed due to their lognormal distribution. In order to obtain eddy-induced Chl-a anomalies (Chl-a'), the satellite log-transformed Chl-a field was first filtered with a 7-90 day Butterworth time filter. The time-filtered Chl-a field was then anti-log-transformed to get the original units of mg/m$^3$ for direct comparisons of their results inside eddies (Gaube et al., 2013). Finally, a 600 km high-pass spatial filter was applied to the time-filtered Chl-a field, generating an eddy-induced Chl-a' field.

### 2.1.2 In-situ Data

The study utilizes Argo profiles to construct the subsurface eddy structures. The Argo floats provide temperature and salinity measurements from the sea surface to thousands of meters below, allowing for a comprehensive understanding of subsurface conditions. Besides, the daily climatology of subsurface temperature and salinity values is acquired from the CSIRO Atlas of Regional Seas 2009 (CARS09) product. These climatological values are then subtracted from the Argo profiles, enabling the isolation of anomalies specific to the eddy features. In addition, we used the density-based mixed layer depth (MLD) data derived from Argo floats by Holte and Talley (2009) to study the relationship between MLD variations and "abnormal" eddies. MLD data within 1.5 radii (R) of the eddy core on the same day were selected for the study.

Furthermore, to study the differences in vertical chlorophyll distributions between SEs and SSEs, the study utilizes Biogeochemical Argo (BGC-Argo) floats equipped with bio-optical sensors to measure biogeochemical variables. For each BGC-Argo profile, we selected the highest-level data mode (delayed mode), produced later (over 1 year), and required control and validation by a scientific expert. Only profile data flagged as good quality were considered in the study. In addition, we conducted quality control on Chl-a profiles. First, a three-point moving median filter was applied on each profile to remove spikes (Haëntjens et al., 2020; Bisson et al., 2019). Next, we followed the calibration procedure of Roesler et al. (2017) and Haëntjens et al. (2020) to adjust the Chl-a data. Finally, quality control was applied to eddy-collocated BGC-Argo floats using the following criteria: (1) Chl-a data from the upper 10 m were excluded from analyses because large variability and high uncertainty were observed there (Su et al., 2021b); (2) Besides, each profile must contain at least one data point at a depth of 200 m or greater. It is because the Chl-a is generally located at the base of the euphotic layer (50 - 200 m) in the NIO (Mignot et al., 2014); (3) There are more than 5 observations between 10 m and 200 m.

## 2.2 Methods

### 2.2.1 DL-based Eddy Identification Model

The study aims to extract SEs and SSEs based on the differences in their thermodynamic structures. Fig. 1a illustrates the shape of isopycnal levels for SEs and SSEs, as described in the study by Assassi et al. (2016). SAEs exhibit positive SSHA and the deepening of isopycnals, resulting in negative sea surface density (SS$\rho$) anomalies. Conversely, SCEs show negative SSHA and the upward displacement of isopycnals, inducing positive anomalies in SS$\rho$. Therefore, both SCEs and SAEs show negative relationships between SS$\rho$ and SSHA. For SSEs, the scenario is slightly different. Subsurface-intensified AEs

(SSAEs) also exhibit positive SSHA, similar to SAEs. However, the shape of isopycnal levels associated with SSAEs is lens-like, indicating an upward displacement of water above the center and a downward displacement below it. Similarly, subsurface-intensified CEs (SSCEs) maintain negative SSHA, as observed in SCEs. However, the isopycnal levels above SSCEs exhibit a depressed shape, indicating a downward displacement of water, while the isopycnal below the SSCEs display a domed shape, indicating an upward displacement. Consequently, the SS$\rho$ anomalies within SSEs have the opposite sign

compared to SEs, leading to a positive SS$\rho$/SSHA ratio for both SSAEs and SSCEs. Therefore, the sign of SS$\rho$/SSHA can be used as an indicator to distinguish SAEs and SSAEs, or SCEs and SSCEs. However, it is important to note that SS$\rho$ cannot be directly measured from remote sensing observations. Instead, at first order, SS$\rho$ are primarily influenced by SST variations, which can be observed remotely. Thus, the SSTA-SSHA relationship within eddies can be employed to differentiate between SEs and SSEs, successfully applied in previous studies (Trott et al., 2019; Greaser et al., 2020; Wang et al., 2019).

Accordingly, a DL-based model is developed to distinguish between SEs and SSEs by integrating satellite-derived SSHA and SSTA data mentioned in section 2.1.1. As shown in Fig. 1b, the DL-based model employs an encoder-decoder architecture (Ronneberger et al., 2015) for feature extraction from SSHA and SSTA data. The encoder-decoder architecture offers several advantages in terms of simplicity, reduced training time, fewer parameters, and lower sample requirements. Consequently, it effectively reduces computational complexity while efficiently extracting features. In the encoder part of the model,

convolutions are utilized to extract spatial information from the input image, followed by max-pooling to reduce the feature dimensions progressively. In the decoder part, up-convolutions are employed to restore object details and spatial information. Besides, features from the corresponding encoder and decoder layers are concatenated to enrich the decoded information. Especially to address the complex nonlinear relationship between SSHA and SSTA within mesoscale eddies, a dense connection network (Dolz et al., 2018) is incorporated into the encoder part to facilitate the fusion of remote sensing SSHA

and SSTA data. Unlike traditional convolutional neural networks, where information flows sequentially from one layer to the next, the dense connection network establishes direct connections from any layer to all subsequent layers in a forward manner. The forward propagation is represented by Eq. (1):

$$x_l^s = H_l^s([x_{l-1}^1, x_{l-1}^2, x_{l-2}^1, x_{l-2}^2, \cdots, x_0^1, x_0^2]), \tag{1}$$

where $x$ represents a single network layer, the superscript $s$ denotes the modality of the network layer, and the subscript $l$ indicates the layer number. The function $H_l^s$ represents a composite operation that includes batch normalization (BN), rectified linear unit (ReLU), and convolutional operations. By incorporating dense connections, the DL-based model introduces implicit deep supervision, enhancing learning capabilities and improving information flow and gradient throughout the model. It not only facilitates the extraction of correlated spatiotemporal features of SSHA and SSTA at different scales but also mitigates the issue of gradient vanishing that commonly arises with increasing network depth. Consequently, the proposed model ensures a more efficient and accurate training process.

The DL-based eddy identification model was trained and validated using datasets generated by a traditional SSH-based method (Liu et al., 2016; Chelton et al., 2011a), which extracts AEs and CEs by searching closed SSHA contours. Then, to determine whether an AE is SAE or SSAE and a CE is SCE or SSCE, the study calculates the mean SSTA within one radius within eddies. The mean SSTA within SCEs and SSAEs is negative, while positive within SAEs and SSCEs. As a result, we obtained the training dataset consisting of 1827 samples from 2000-2004 and the testing dataset consisting of 365 samples from 2005. Each sample contains four kinds of eddies in the NIO: SSCEs, SCEs, SAEs, and SSAEs, with pixels labeled as '1', '2', '3', and '4', respectively. The study utilized dice loss and categorical accuracy to optimize and estimate the DL-based eddy identification model. The dice loss is defined as:

$$Loss = 1 - Dicecoef(P, G) \tag{2}$$

$Dicecoef(P, G)$, i.e., dice coefficient, a popular cost function for segmentation problems in deep learning. Given the predicted segmentation P and the ground truth region G, the dice coefficient is calculated as:

$$Dicecoef(P, G) = \frac{2|P \cap G|}{|P| + |G|} \tag{3}$$

where |.| is the sum of elements in the area. A good segmentation result is explained by a dice coefficient close to 1. A low dice coefficient (near 0) indicates poor segmentation performance. Categorical accuracy is a metric that calculates the mean accuracy rate across all predictions for multi-class classification problems, which is defined as follows:

$$Categorical\ accuracy = \frac{TP + TN}{TP + TN + FP + FN} \tag{4}$$

TP, TN, FP, and FN represent the number of true positives, true negatives, false positives, and false negatives, respectively. When the model was evaluated on the testing samples, it achieved a loss of approximately 0.12 and an accuracy of around 0.95 (Figs. 1c-d). With a low loss value and a high accuracy rate, the DL-based model demonstrated promising results in accurately identifying and classifying the different types of eddies in the testing samples (Fig. 1e). Considering the resolution and precision of the SSHA product (Pujol et al., 2016), individual eddies with amplitudes ≥ 2 cm and radii ≥ 35 km are selected to avoid the noises from low-energy eddies in the study.

## 2.2.2 Surface and Subsurface Composite Analysis over Eddies

The study conducted a surface composite analysis combining eddy-induced SSTA and Chl-a' data on a normalized grid. The analysis aims to examine the composite patterns of SSTA and Chl-a' associated with different types of eddies. The eddy-induced SSTA and Chl-a' values within a region twice the radius (R) of each eddy were collected to construct the surface composite analysis. These values were then interpolated onto a normalized circle of the same size, as depicted in Fig. 2a. Next, composite SSTA and Chl-a' maps were generated by averaging the normalized anomaly fields over the eddies of the same type. This process involved grouping the eddies based on their characteristics and calculating the average SSTA and Chl-a' values at each grid point within the normalized circle for each group of eddies.

To analyze the characteristics of eddies' subsurface structures, we select Argo profiles co-located within 1.5 R from the eddy core to construct the 3D structure of mesoscale eddies. Quality control was first applied to eddy-collocated Argo floats using the following criteria: (1) Only profiles data flagged as good quality were considered; (2) each Argo profile must contain a data point at a depth of 10 m or less, and at least one data point at a depth of 1,000 m or greater; (3) there are more than 30 observations between 0 m and 1,000 m. Secondly, temperature and salinity data were interpolated on a regular 10 m grid ranging from 10 m to 1,000 m  because Argo floats may or may not have observed data at the surface. Thirdly, the Argo profiles were processed by subtracting the CARS09 dataset to obtain temperature and salinity anomalies, specifically within the eddy regions. Moreover, potential density anomalies were calculated by temperature and salinity anomalies according to the International Thermodynamic Equation of Seawater (Mcdougall and Barker, 2011). Subsequently, the temperature and potential density anomalies within 1.5R of mesoscale eddies were interpolated into $0.1R \times 0.1R$ grid points up to a horizontal distance of 1.5R (Fig. 2b) by the inversed distance weighting interpolation method (Bartier and Keller, 1996) at each depth level (Sun et al., 2019; Yang et al., 2013; Dong et al., 2017). For each grid point, Argo profiles located within the horizontal range of 0.1R are set the weight value:

$$w_i = e^{-(\frac{d}{R})^2} \tag{5}$$

where d denotes the distance from the profile to the grid point. The final temperature or potential value at each grid point, $N_{grid}$, is calculated from the profile values $N_i$ as:

$$N_{grid} = \frac{\sum w_i N_i}{\sum w_i} \tag{6}$$

## 3 Results

### 3.1 Case Studies of SST and Chl-a within SEs and SSEs

The study conducts case studies to preliminarily examine characteristics of SSTA and Chl-a' within SEs and SSEs. As shown in Fig. 3a, the DL-based model detected an SAE and an SCE on the AS's west coast on February 2, 2005. The SAE displays

positive signatures in SSTA images, indicating warm water, and negative signatures in Chl-a' images, indicating lower Chl-a concentrations. In contrast, the SCE shows negative SSTA and positive Chl-a' signatures. These findings are consistent with conventional knowledge, where AEs are generally identified as warm rings with lower Chl-a concentrations in ocean color maps, while CEs exhibit the opposite pattern (Gaube et al., 2014; Hsu et al., 2016). Fig. 3b shows an example of an SSAE on the east coast of the AS on March 13, 2002. The SSAE is associated with cold water and displays positive Chl-a' signatures, indicating higher Chl-a concentrations. Similarly, Fig. 3c presents an example of an SSCE in the North Central BoB on November 28, 2014. The SSCE is associated with positive SSTA, indicating warm water, but exhibits negative Chl-a' values, indicating lower chlorophyll-a concentrations. The above findings suggest that SSEs exhibit distinct effects on Chl-a concentrations compared to SEs.

## 3.2 Spatial Distribution of SST and Chl-a within SEs and SSEs

The study applied the DL-based model to identify SEs and SSEs in the NIO from 2000 to 2015. As a result, 61,095 SAEs, 38,889 SSAEs, 70,596 SCEs, and 46, 294 SSCEs are observed. The number represents the aggregate count of eddies of identical type across all eddy snapshots during 2000-2015. Figs. 4a-d depict the spatial distribution of eddy concentration, representing eddy numbers of the same type observed within a 1°×1° grid during 2000-2015. In the NIO, the number of SEs (SAEs and SCEs) accounted for 61% of the total, while SSEs (SSAEs and SSCEs) constituted 39%.

The coastal areas of the Arabian Peninsula and the East Indian Coastal Current (EICC) in the BoB exhibited a pronounced abundance and prevalence of SEs and SSEs. These regions are known for their active eddy generation mechanisms, including coastal upwelling, Rossby waves, and barotropic instabilities in the AS (Trott et al., 2018; Zhan et al., 2020), as well as monsoon conversion, EICC instability, and westward Rossby wave energy transmission in the BoB (Somayajulu et al., 2003; Chen et al., 2012; Cheng et al., 2018; Cui et al., 2016). Figs. 4e-h display the spatial distributions of eddy-induced SSTA averaged within a 1°×1° grid. SAEs and SSCEs exhibit positive SSTA values (Figs. 4e, 4h), indicating warmer water, while SSAEs and SCEs display negative SSTA values (Figs. 4f, 4g), indicating cooler water. The distinct SSTA signatures exhibited by these eddies align with the expected patterns associated with SEs and SSEs defined in section 2.2.1.

Fig. 5 illustrates the spatial distribution of Chl-a' averaged within a 1°×1° grid, specifically induced by SEs and SSEs in the NIO during 2000-2015. Chl-a' induced by SAEs (Fig. 5a) exhibits predominantly negative values across most areas of both basins. The western parts of both basins, particularly in the Somali Current (SC) region in the AS, exhibit the lowest concentrations of Chl-a'. It suggests that SAEs are associated with decreased phytoplankton biomass or lower productivity in these regions. Whereas, Chl-a' induced by SSAEs (Fig. 5b) shows predominantly positive signals in more areas, with a concentration observed along the northeastern coasts of both basins, which indicates that SSAEs are associated with higher productivity in these regions. For SCEs (Fig. 5c), eddy-induced Chl-a' exhibits positive values and a higher concentration along the SC region. In contrast, in most areas, Chl-a' induced by SSCEs (Fig. 5d) is generally insignificant. It shows negative values in the Gulf of Aden and north of the Andaman Sea. It implies that SSCEs may have less effect on primary productivity in these regions than SCEs.

## 3.3 Seasonal Variations of Composite SST and Chl-a within SEs and SSEs

Considering distinct monsoon and productivity backgrounds in the AS and the BoB regions, we conducted a composite analysis of SSTA and Chl-a' within SAEs, SSAEs, SCEs, and SSCEs in summer and winter monsoons for both basins. Fig. 6 shows composite SSTA over SEs and SSEs in the AS and the BoB during summer and winter monsoons. In both basins, composite SSTA over the SEs and SSEs exhibit similar monopole patterns with opposite signals. Specifically, the composite SSTA signals for SAEs were positive, while those for SCEs were negative. Conversely, the signals for SSAEs were positive, and SSCEs displayed negative SSTA patterns. Despite the opposite SSTA signals between the SEs and SSEs, their magnitudes were comparable within the same season, indicating that the inversed SSTA signal within SSEs should not be overlooked.

In addition, eddy-induced SSTA over both SEs and SSEs are more pronounced during summer compared to winter in the AS (Figs. 6a-h). Table 1 shows that composite SSTA extremums within SEs and SSEs during summer are at least 1.6 times higher than those observed during winter. The seasonal variation in the intensity of monsoon winds is suggested to influence the impact of eddy-induced SSTA in the AS throughout the year. The intensified southwesterly winds during the summer monsoon contribute to enhanced upwelling and mixing processes, leading to greater changes in SSTA induced by eddies. In contrast, the weaker northeasterly winds during the winter monsoon are associated with reduced upwelling and mixing, leading to relatively less pronounced eddy-induced SSTA. However, composite SSTA over the SEs and SSEs did not exhibit a significant seasonal variation in the BoB. The intensities of eddy-induced SSTA were slightly larger during the summer monsoon than in winter, with a difference of 0.01°C (Table 1). The observed slight difference in intensity of composite eddy-induced SSTA between the BoB and the AS can be primarily attributed to the seasonal variations in monsoon winds. The BoB exhibits a less pronounced seasonal variation in monsoon winds than the AS. During the summer monsoon, the AS experiences stronger winds than the BoB, while both basins encounter relatively weaker winds during the winter monsoon. The divergence in wind strength contributes significantly to the distinct intensity of eddy-induced SSTA between the two basins.

Despite the opposing signals of SSTA induced by SEs and SSEs, they generally exhibit a consistent signal in terms of Chl-a' (Fig. 7). In the AS, composite Chl-a' shows dipole patterns with positive signals for SAEs and SSAEs and negative signals for SCEs and SSCEs (Figs. 7a-h). Although the Chl-a' signals within the SEs and SSEs exhibit similar patterns, their magnitudes significantly differ. According to the data presented in Table 1, the Chl-a' induced by SAEs during summer and winter are -0.040 mg/m$^3$ and -0.049 mg/m$^3$, respectively. Conversely, the Chl-a' induced by SSAEs during summer and winter are -0.017 mg/m$^3$ and -0.012 mg/m$^3$, respectively. It indicates that the Chl-a concentration within SAEs is notably lower than SSAEs, with approximately half of the concentration observed in the latter. On the other hand, the Chl-a concentration within SCEs is two to three times higher compared to SSCEs. Specifically, the Chl-a' induced by SCEs during summer and winter is 0.076 mg/m$^3$ and 0.084 mg/m$^3$, respectively, whereas the Chl-a' induced by SSCEs during summer and winter is 0.018 mg/m$^3$ and 0.039 mg/m$^3$, respectively. Notably, Chl-a' intensities over both SEs and SSEs in the AS demonstrate a relatively consistent pattern between the summer and winter monsoons, with no significant variation observed. Winter productivity in the AS has

been suggested to be comparable to, or occasionally even surpass, that of the summer (Piontkovski et al., 2011). The enhanced productivity during winter is attributed to the convective winter mixing, which facilitates the upward transport of nutrients to the surface layer (Banse and English, 2000).

However, significant seasonal variations are observed in the impact of SEs and SSEs on Chl-a concentration in the BoB (Figs. 7i-p). During the summer monsoon, eddy-induced Chl-a' over the SEs and SSEs exhibit similar patterns (Figs. 7i-l), with slight differences in magnitudes. As shown in Table 1, the Chl-a' induced by SAEs and SSAEs are -0.029 mg/m$^3$ and -0.021 mg/m$^3$, indicating a decrease in chlorophyll concentration compared to the surrounding areas. On the other hand, SCEs and SSCEs exhibit positive Chl-a' values of 0.021 mg/m$^3$ and 0.018 mg/m$^3$, respectively, indicating an increase in chlorophyll concentration. During the winter monsoon, composite Chl-a' induced by SEs and SSEs exhibit distinct patterns (Figs. 7m-p). Specifically, SAEs exhibit a predominant presence of negative Chl-a' values, with a minimum concentration of -0.018 mg/m$^3$. In contrast, SSAEs are characterized by positive Chl-a' values, reaching a maximum concentration of 0.027 mg/m$^3$. Besides, SCEs predominantly exhibit positive Chl-a' values, with a maximum concentration of 0.033 mg/m$^3$, which is approximately three times higher than that induced by SSCEs. Additionally, the concentration of eddy-induced Chl-a' in the BoB was considerably lower than in the AS. The lower Chl-a concentration within eddies in the BoB is attributed to weakened vertical mixing resulting from freshwater-induced stratification and relatively weaker winds (Prasanna Kumar et al., 2002).

## 4 Discussion

Relying solely on the SSHA-SSTA relationship may lead to potential misidentification of SSEs due to various sources of errors (Assassi et al., 2016). For example, it is challenging when dealing with eddies exhibiting multicore structures of similar strength, making it difficult to determine the location of the most intense core accurately. Besides, in regions where salinity plays a significant role in stratification, variations of SS$\rho$ may not be dominated by SST variations at first order. In order to validate the accuracy and robustness of the DL-based eddy identification model, the study employs quality-controlled Argo profiles to construct subsurface eddy structures for both SEs and SSEs in the AS and the BoB. During 2000-2015, the numbers of Argo profiles within SAEs, SSAEs, SCEs, and SSCEs were as follows: 2777, 1028, 2336, and 1747 profiles in the AS, and 778, 374, 648, and 424 profiles in the BoB, respectively.

Fig. 8 provides insights into the subsurface temperature anomalies within SEs (SAEs and SCEs) and SSEs (SSAEs and SSCEs) in the AS and the BoB. In the AS, SAEs and SCEs exhibit positive and negative temperature anomalies throughout the structure, with maximum and minimum temperature anomalies at approximately 100 m (Figs. 8a, c). Conversely, SSAEs and SSCEs display negative and positive temperature anomalies approximately within the MLD at around 30 m, contrasting with their subsurface layers (Figs. 8b, d). Similar differences in the subsurface temperature structure between SEs and SSEs are also observed in the BoB (Figs. 8e-h). Specifically, the SAEs (SCEs) showed positive (negative) temperature anomalies throughout the water column, while the SSAEs (SSCEs) displayed a small cap of cold (warm) water within the MLD.

Furthermore, the study constructs vertical structures of potential density within the eddies (Fig. 9) to determine whether the
310 isopycnal displacements of SEs and SSEs align with the definition proposed by Assassi et al. (2016). In the AS, SAEs and
SCEs exhibit negative and positive potential density anomalies throughout the structure, respectively (Figs. 9a, c). However,
SSAEs and SSCEs show a small cap of positive and negative potential density anomalies within the MLD, contrasting with
their subsurface layers (Figs. 9b, d). Similar patterns are observed in the BoB, where SSAEs and SSCEs display positive and
negative potential density anomalies within the MLD, respectively (Fig. 9f). Thus, SSAEs generally exhibit positive potential
density anomalies in the near-surface layer, which can be attributed to the upward displacement of isopycnals. In contrast,
SSCEs show negative potential density anomalies due to downward displacement. These findings align well with the schematic
diagram of isopycnal displacements of SEs and SSEs depicted in Fig. 1a. By reconstructing the subsurface structure of eddies,
the study confirms the accuracy of the DL-based model in distinguishing between SEs and SSEs. Besides, Figs. 8-9 reveal that
the difference in the subsurface structure between SEs and SSEs is largely confined to the MLD. Such a result indicates that
the formation of SSEs is dominated by eddy–wind interaction (Mcgillicuddy, 2015), which leads to lens-shaped disturbances
in the thermocline. The relative motion between surface winds and eddy surface currents leads to anomalous Ekman upwelling
(downwelling) within AEs (CEs), which can induce doming (depressing) of the upper ocean density surfaces inside AEs (CEs)
(Gaube et al., 2015).

Additionally, the study reveals subsurface Chl-a characteristics of SEs and SSEs using eddy-collocated BGC-Argo floats.
In the NIO, spanning the years 2000 to 2015, we identified a total of 30 BGC-Argo profiles located within 1.5R of AEs (CEs),
which met our rigorous quality control criteria, as detailed in section 2.1.2. Among these profiles, 18 (12) BGC-Argo profiles
were found within 1.5R of SAEs (SSAEs), while 32 (13) BGC-Argo profiles were found within 1.5R of SCEs (SSCEs). Despite
the relatively limited number of BGC-Argo profiles, our analysis unmistakably reveals discernible distinctions in the Chl-a
profiles between SAEs and SSAEs, as well as SCEs and SSCEs. As shown in Fig. 10, the variations in Chl-a induced by eddies
are predominantly concentrated within the upper 100 m of the water column. The observation aligns with previous research
findings, which suggest that Chl-a tends to be concentrated at the base of the euphotic layer, typically spanning depths of 50
to 200 m in the NIO (Mignot et al., 2014). Furthermore, it's worth noting that Chl-a levels induced by SSAEs exhibit a
substantial increase, approximately 0.1 mg/m$^3$, compared to those induced by SAEs within the upper 30 m (Fig. 10a). In
contrast, the Chl-a concentrations induced by SSCEs are notably lower, approximately 0.1 mg/m$^3$, in comparison to SCEs
within the upper 50 m (Fig. 10b). These disparities can be attributed to distinct displacements of isopycnals between SEs and
SSEs. The convex of isopycnals within SSAEs leads to the ascent of deeper water to the surface layer. This process facilitates
the vertical transport of nutrients, promoting enhanced biological productivity and higher concentrations of Chl-a within
SSAEs than SAEs. The vertical movement of water masses and the associated nutrient supply contribute to the favorable
conditions for phytoplankton growth and the accumulation of Chl-a in SSAEs. Similarly, the concave of isopycnals within
340 SSCEs leads to the subduction of surface water, resulting in lower Chl-a concentrations compared to SCEs.

**5 Conclusions**

The study proposes a DL-based model that integrates satellite-derived SSH and SST data to accurately distinguish between SEs and SSEs in the NIO during 2000-2015. In the NIO, the number of SEs (SAEs and SCEs) accounted for 61% of the total, while SSEs (SSAEs and SSCEs) constituted 39%. SAEs and SCEs exhibit positive and negative SSTA, contrary to SSAEs and SSCEs, respectively. In addition, SEs and SSEs show significant differences in spatial characteristics and composite patterns of eddy-induced Chl-a. On the one hand, SAEs (SCEs) induce negative (positive) anomalies in Chl-a concentration, with the most significant effects observed in the Somali Current region. However, SSAEs cause positive Chl-a anomalies along the northeastern coast of both basins, while SSCEs lead to negative Chl-a anomalies in the Gulf of Aden and the northern part of the Andaman Sea. On the other hand, composite Chl-a within SAEs is considerably lower compared to SSAEs, which is about twice times lower in the latter. In contrast, the Chl-a concentration in SCEs is twice or three times higher than in the SSCEs. Moreover, using BGC-Argo profiles, SEs and SSEs show significant differences in subsurface Chl-a distribution. Chl-a induced by SSAEs is ~ 0.1 mg/m$^3$ greater than that caused by SAEs in the upper 30 m, while Chl-a induced by SSCEs is ~ 0.1 mg/m$^3$ less than that caused by SCEs in the upper 50 m.

The distinct subsurface structures between SEs and SSEs provide insight into the contrasting impacts on Chl-a distribution. SAEs and SCEs exhibit negative and positive potential density anomalies throughout the structure. However, SSAEs exhibit positive potential density anomalies within the MLD, which can be attributed to the upward displacement of isopycnals. The upward movement facilitated the transport of deeper water to the surface layer, inducing higher Chl-a concentrations within SSAEs. Besides, SSCEs show negative potential density anomalies above the MLD due to the downward displacement of isopycnals, leading to lower Chl-a concentrations than SCEs. In conclusion, the study demonstrates the effectiveness of the DL-based model in distinguishing between SEs and SSEs by fusing remote sensing SSH and SST data. By applying the model, the study enhances the comprehension of the impacts of SSEs on Chl-a distribution and contributes to a deeper understanding of the complex interactions between eddy dynamics and biogeochemical processes.

**Data availability**

All data used in the analysis are available in public repositories. The SSHA dataset is available at https://doi.org/10.48670/moi-00148. OISST product can be downloaded from https://psl.noaa.gov/data/gridded/data.noaa.oisst.v2.highres.html. The Chl-a data were downloaded from http://www.globcolour.info. Argo data can be downloaded from http://www.coriolis.eu.org. CARS2009 data was obtained from http://www.marine.csiro.au/~dunn/cars2009/. BGC-Argo data can be downloaded from https://dataselection.euro-argo.eu/. The mixed layer depth data can be downloaded from http://mixedlayer.ucsd.edu/. The code of the DL-based model is available at http://dx.doi.org/10.6084/m9.figshare.23599683, and the dataset of eddy-induced chlorophyll-a used in this paper can be downloaded from http://dx.doi.org/10.6084/m9.figshare.23599473.

**Author contributions**

Yingjie Liu and Xiaofeng Li: writing, analysis, and revision. All authors provided feedback on the analysis and interpretation of results and contributed to reviewing and editing the manuscript. All authors have read and agreed to the published version of the manuscript.

**Competing interests**

The authors declare that they have no conflict of interest.

**Acknowledgments**

This work was supported by the Qingdao National Laboratory for Marine Science and Technology, the special fund of Shandong province (No. LSKJ202204302), the Natural Science Foundation of Shandong Province (ZR2020MD083), the National Natural Science Foundation of China (U2006211 and 42306194), the Strategic Priority Research Program of the Chinese Academy of Sciences (XDA19060101 and XDB42000000), Major scientific and technological innovation projects of Shandong Province (2019JZZY010102), and the CAS Program (Y9KY04101L).

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

**Table 1.** Composite extremum values ±1 Confidence Interval (CI) for SSTA/Chl-a' over four kinds of eddies. The CI was computed at the location of SSTA/Chl-a' extremum in composite maps.

| | | AS | | BoB | |
|---|---|---|---|---|---|
| | | Summer | Winter | Summer | Winter |
| SSTA (°C) | SAEs | 0.14±0.004 | 0.08±0.002 | 0.09±0.003 | 0.08±0.003 |
| | SSAEs | -0.15±0.004 | -0.09±0.002 | -0.12±0.004 | -0.11±0.003 |
| | SCEs | -0.16±0.004 | -0.10±0.002 | -0.11±0.004 | -0.10±0.003 |
| | SSCEs | 0.16±0.004 | 0.07±0.002 | 0.11±0.003 | 0.10±0.003 |
| Chl-a' (mg/m$^3$) | SAEs | -0.040±0.004 | -0.049±0.005 | -0.029±0.004 | -0.018±0.002 |
| | SSAEs | -0.017±0.004 | -0.012±0.002 | -0.021±0.003 | 0.027±0.004 |
| | SCEs | 0.076±0.007 | 0.084±0.005 | 0.021±0.006 | 0.033±0.004 |
| | SSCEs | 0.018±0.003 | 0.039±0.007 | 0.018±0.005 | 0.010±0.003 |

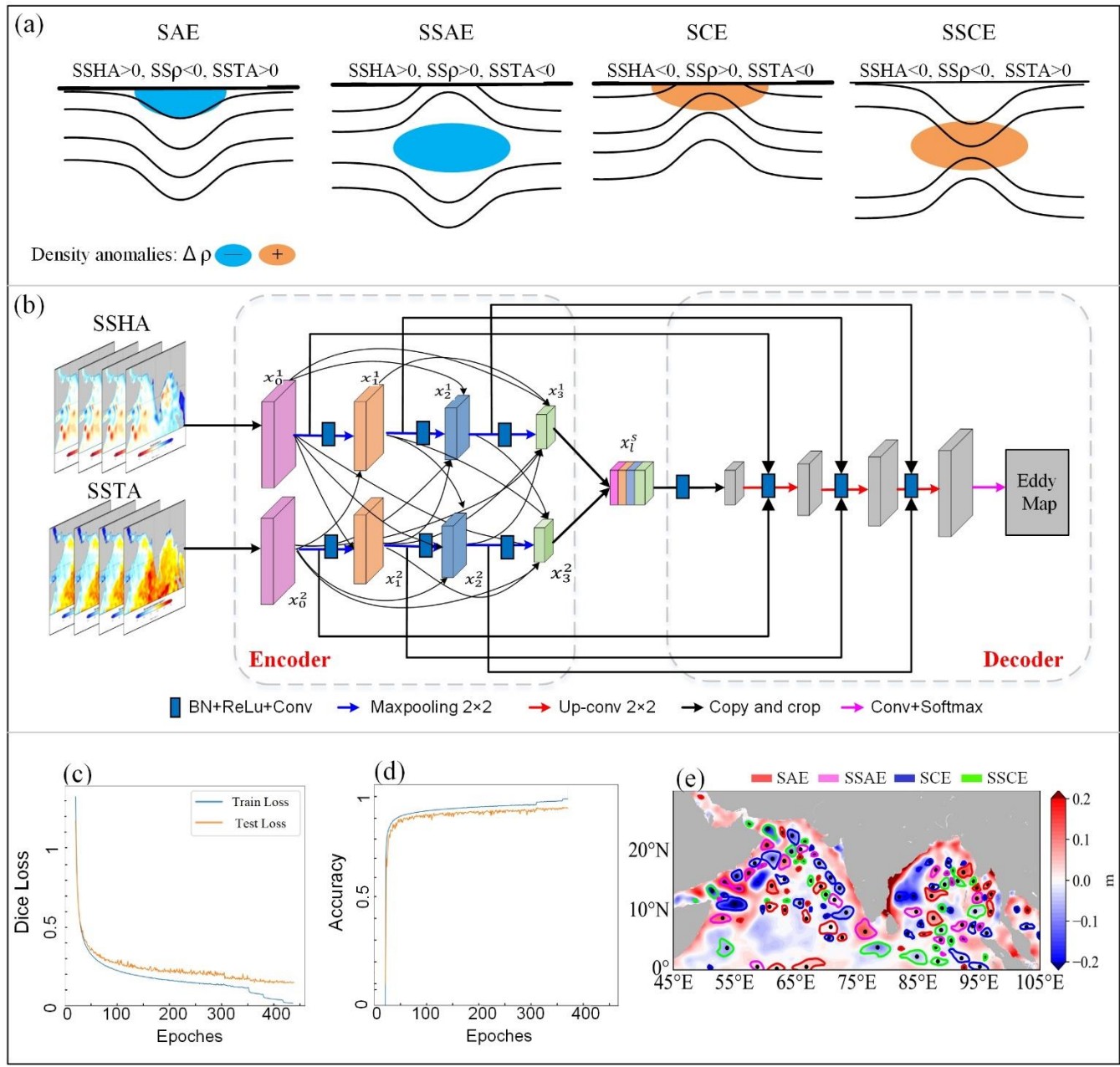

**Figure 1.** (a) Isopycnal displacements, SSHA, and SSTA for SAE, SSAE, SCE, and SSCE. (b) Flow chart of the DL-based eddy identification model. Loss (c) and accuracy (d) curves produced by the DL-based eddy identification model. (e) SSCEs, SCEs, SAEs, and SSAEs detected by the DL-based model on December 1, 2005.

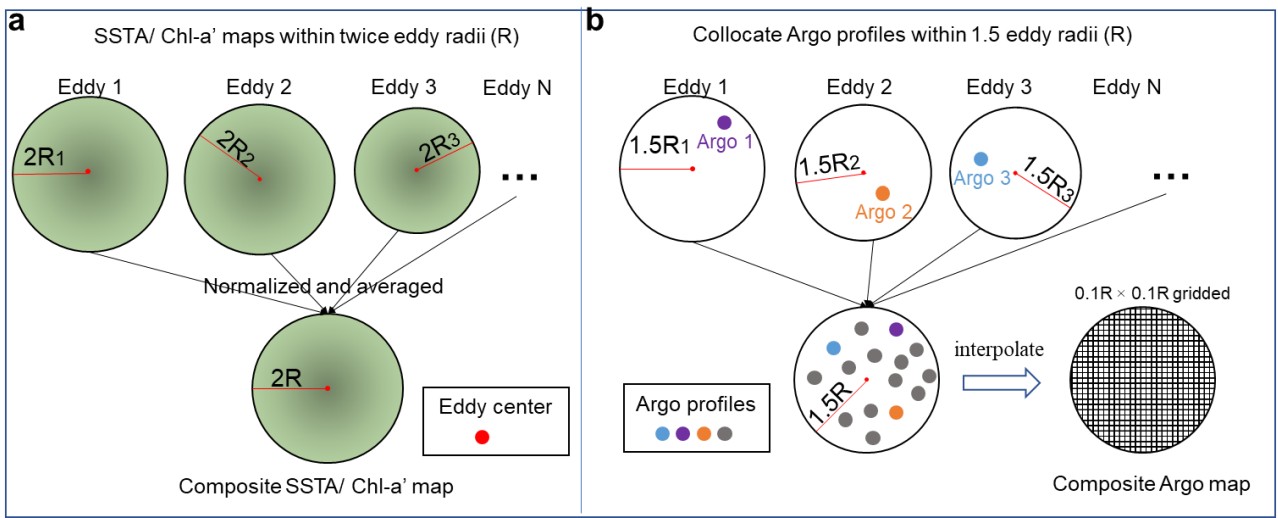

**Figure 2.** Schematic of composite analysis of SST, Chl-a (a), and Argo profiles (b) for eddies.

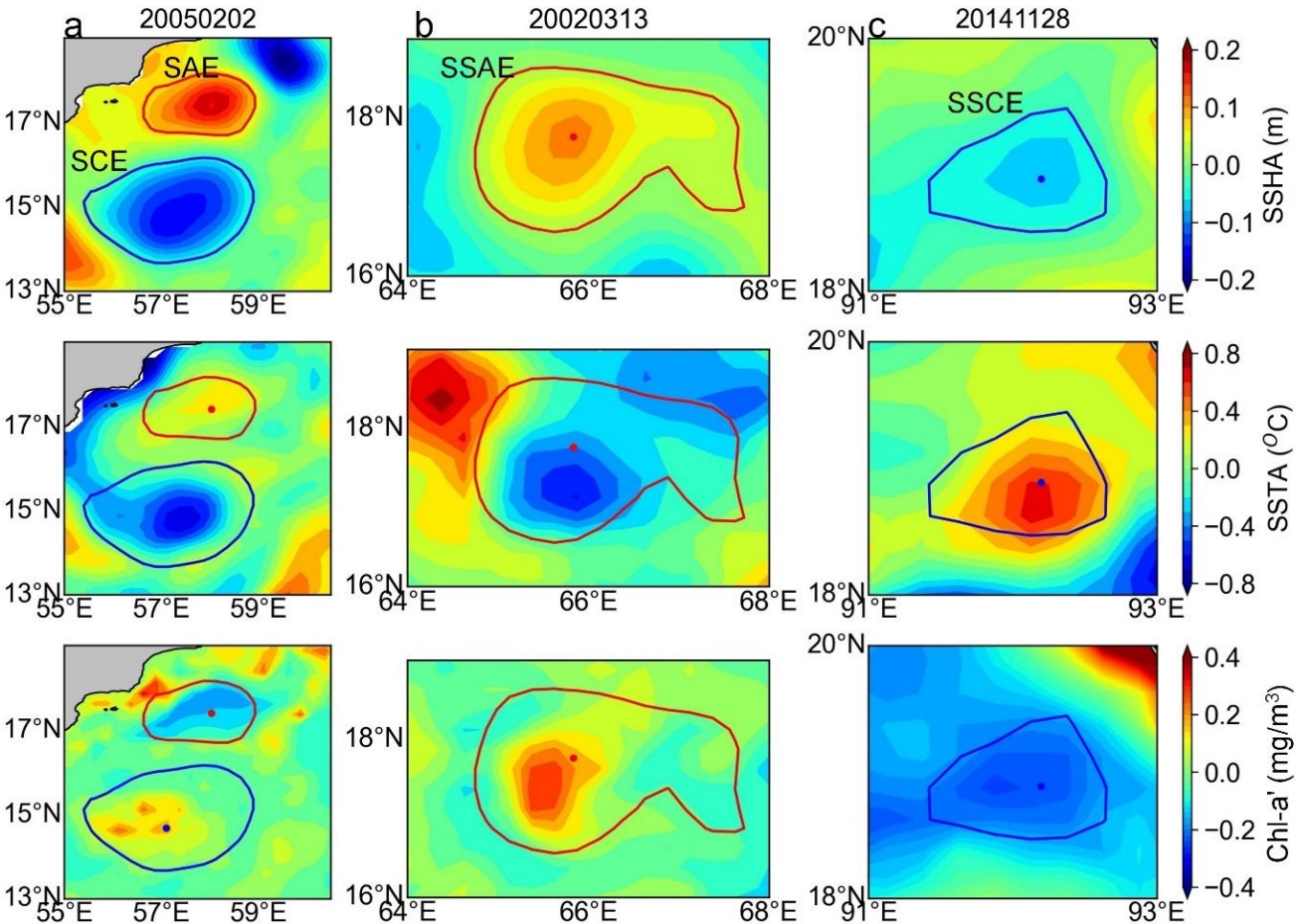

**Figure 3.** Case study of eddy imprints on SSHA, SSTA, and Chl-a' maps for an SAE and an SCE (a), an SSAE (b), and an SSCE (c). Red and blue lines denote eddy edges.

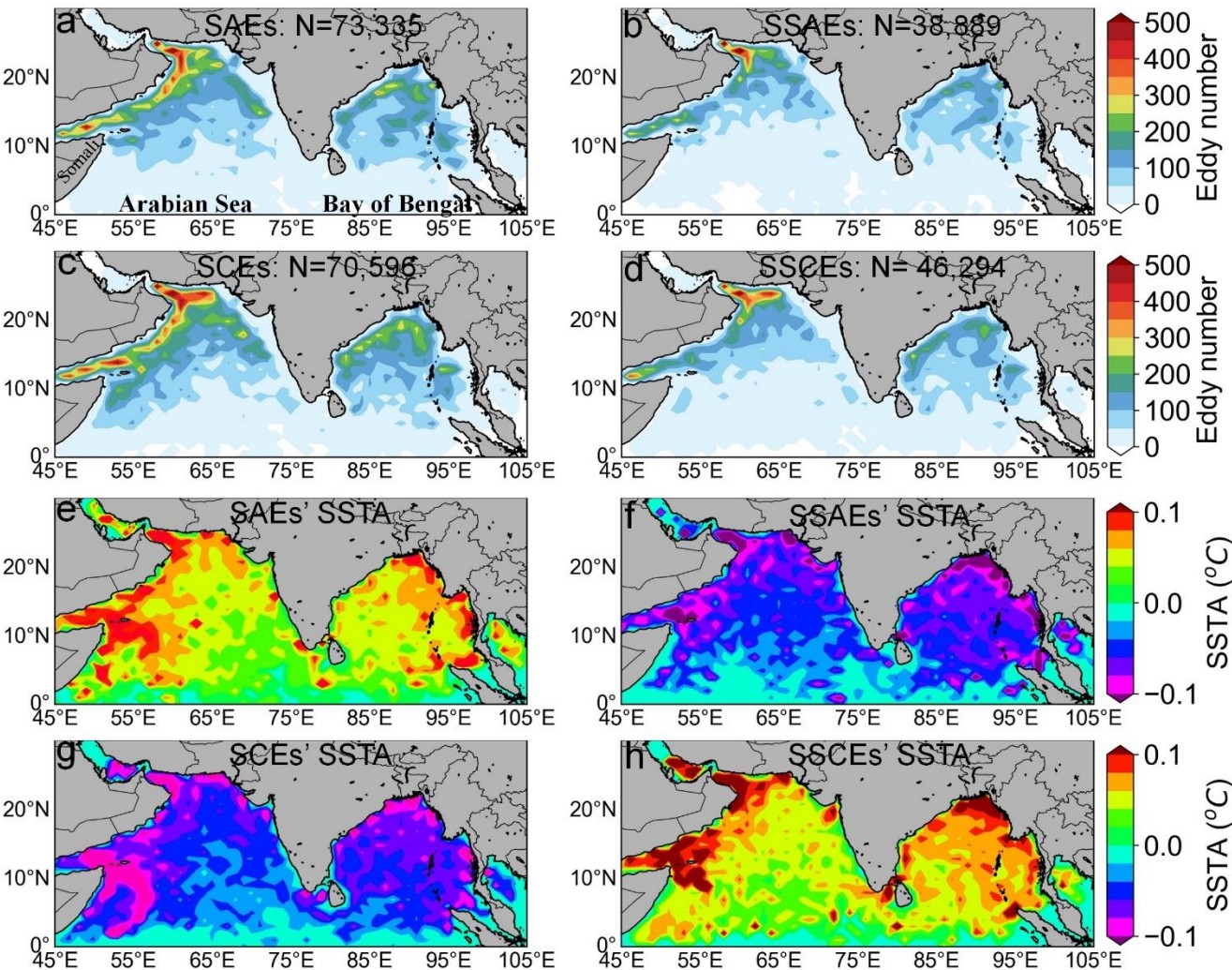

**Figure 4.** The spatial distribution of eddy concentration (a-d) and SSTA (e-h) within SAEs, SSAEs, SCEs, and SSCEs. N is the sum of eddies of the same kind observed in the NIO during 2000-2015.

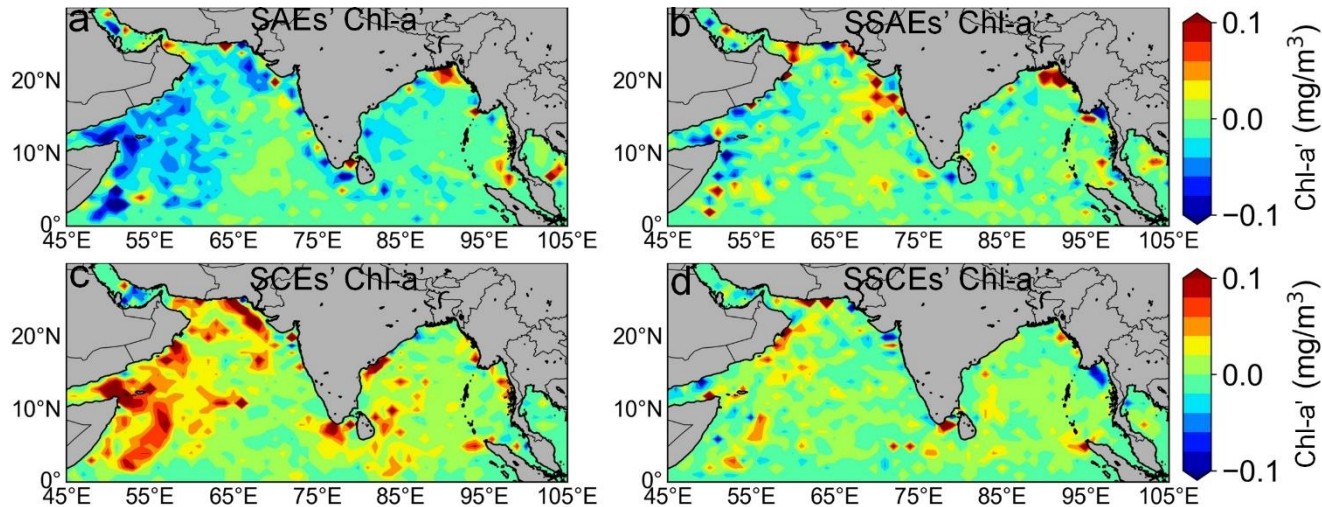

**Figure 5.** Spatial distribution of eddy-induced Chl-a' for SAEs (a), SSAEs (b), SCEs (c), and SSCEs (d) in the NIO during 2000-2015.

570

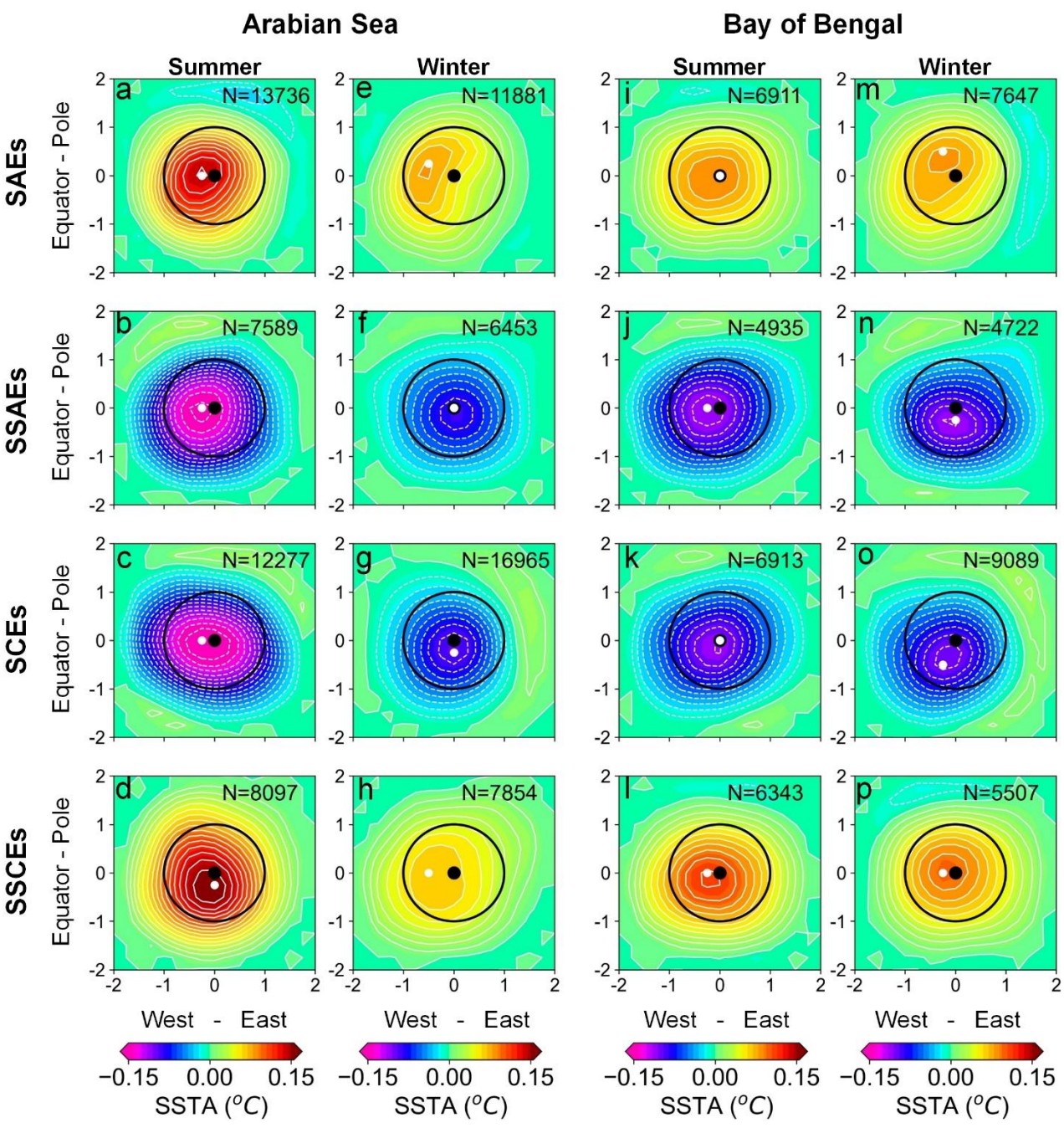

**Figure 6.** Composite SSTA over SAEs, SSAEs, SCEs, and SSCEs in the AS (a-d, i-l) and BoB (e-h, m-p), respectively. Black points denote eddy centers, while white points represent SSTA extremum locations. N is the sum of eddies of the same kind during 2000-2015.

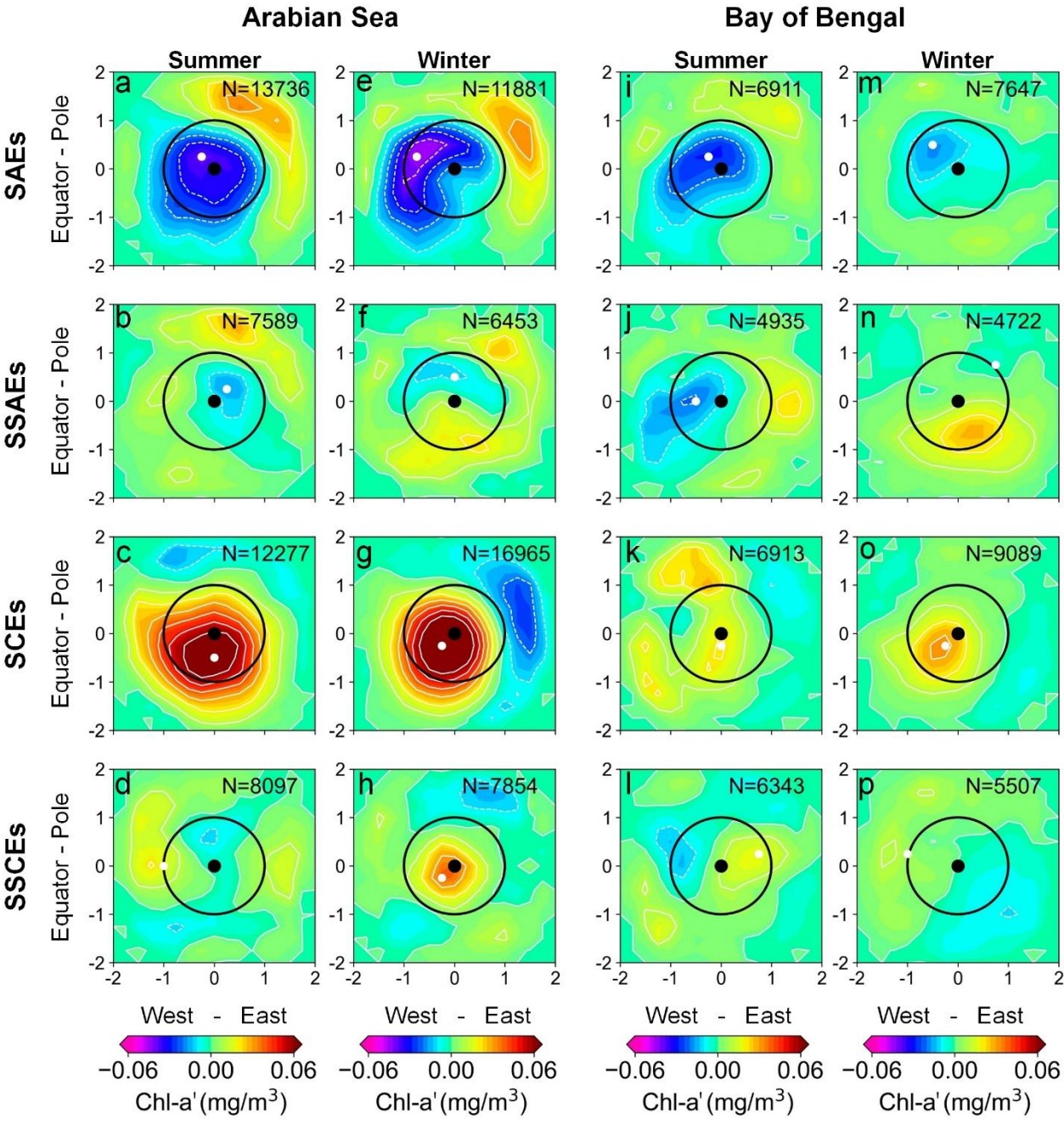

**Figure 7.** Composite Chl-a' over SAEs, SSAEs, SCEs, and SSCEs in the AS (a-d, i-l) and BoB (e-h, m-p), respectively. Black points denote eddy centers, while white points represent Chl-a' extremum locations. N is the sum of eddies of the same kind during 2000-2015.

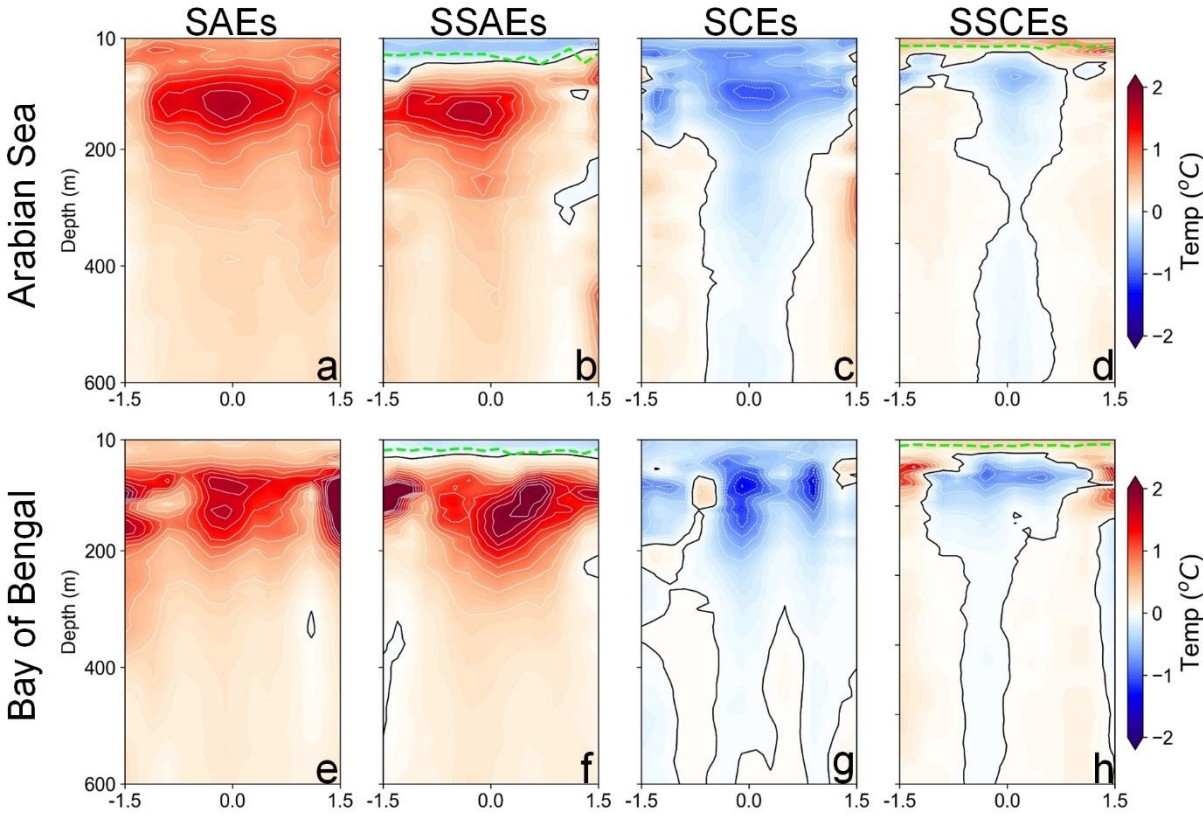

580

**Figure 8.** Composite zonal sections of the vertical temperature structure within SAEs, SSAEs, SCEs, and SSCEs in the AS (a-d) and the BoB (e-h) during 2000-2015. Black lines denote contours in 0°C. The lime dashed lines in SSAEs and SSCEs denote the MLD.

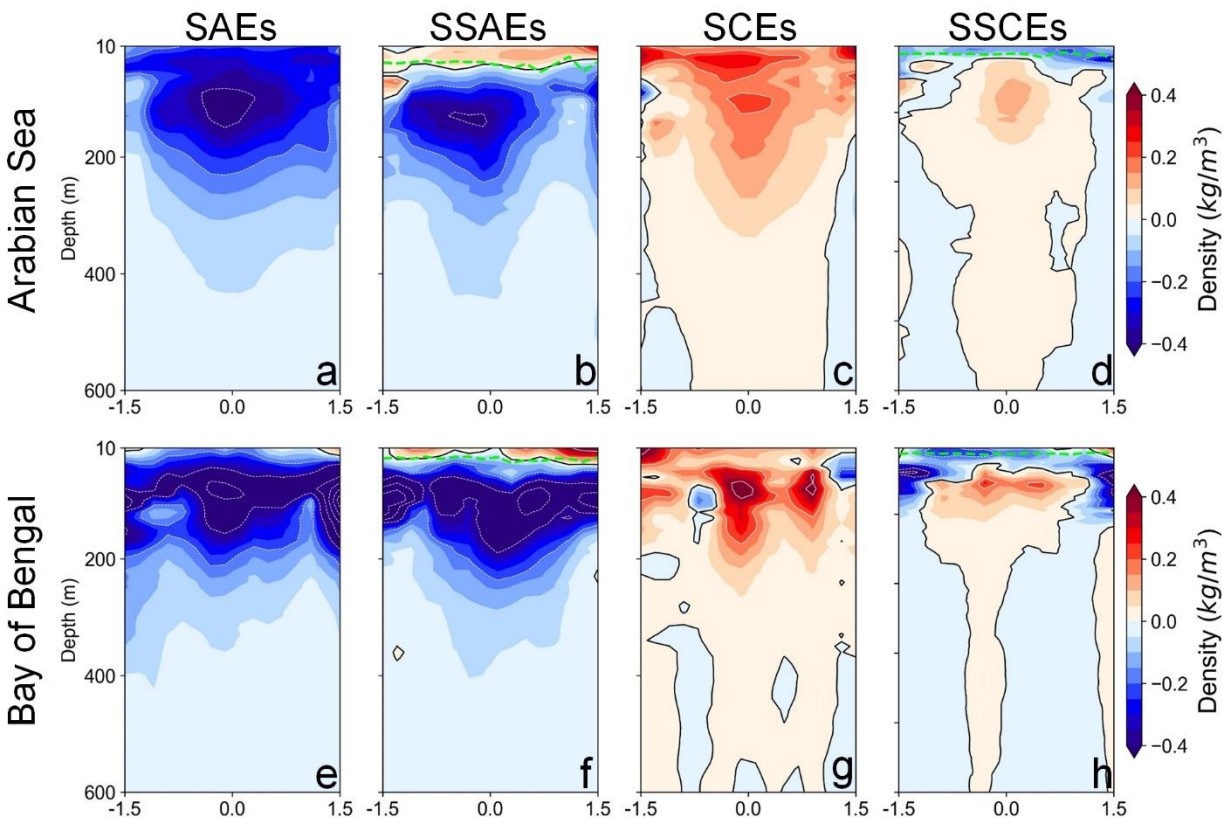

**Figure 9.** Composite zonal sections of the vertical potential density structure within SAEs, SSAEs, SCEs, and SSCEs in the AS (a-d) and the BoB (e-h) from 2000-2015. Black lines denote potential density in 0 kg/m³. The lime dashed lines in SSAEs and SSCEs denote the MLD.

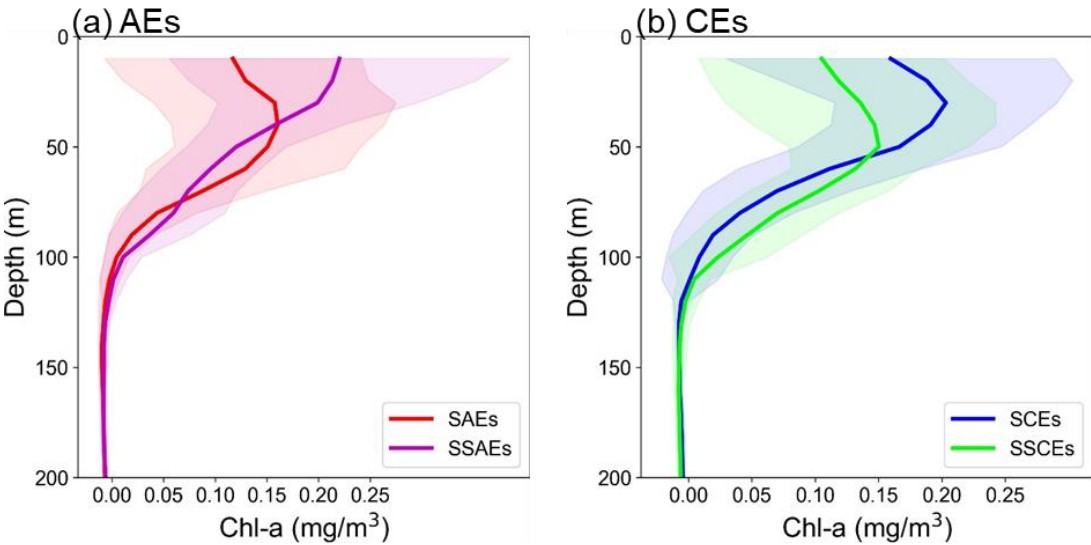

**Figure 10.** Mean (solid line) and standard deviation (shadow) values of BGC-Argo Chl-a profiles for SAEs and SSAEs (a), and SCEs and SSCEs (b) in the North Indian Ocean during 2000-2015.

590