# Peer review of "Impact of Surface and Subsurface-Intensified Eddies on Sea Surface Temperature and Chlorophyll-a in the North Indian Ocean Utilizing Deep Learning"

_EGUsphere, 2023_

## Author Comment (AC1)

1. Consider adding information on the formation mechanism of SSEs in the regions to provide readers with a more comprehensive understanding of SSEs. What's the dominant mechanism of SSE generating in NIO? What cause the different chlorophyll features between SEs and SSEs? If necessary, please give explanations with Argo/BGC-Argo results.

**Response:** Thanks for the suggestion. Firstly, we have incorporated relevant information about the formation mechanisms of subsurface Submesoscale Eddies (SSEs) in the introduction, aiming to provide readers with a comprehensive understanding: "The mechanisms behind the emergence of SSEs are hypothesized to stem from eddy–wind interaction, localized adiabatic processes, barotropic and baroclinic instabilities, or topographic influences (Badin et al., 2011; McGillicuddy, 2015; Meunier et al., 2018; Thomas, 2008)."

Next, in Section 4 of the study, Figs. 8-9 reveal that the difference in the subsurface structure between SEs and SSEs is largely confined to the MLD. Such a result indicates that the formation of SSEs is dominated by eddy–wind interaction (McGillicuddy, 2015), which leads to lens-shaped disturbances in the thermocline. The relative motion between surface winds and eddy surface currents leads to anomalous Ekman upwelling (downwelling) within AEs (CEs), which can induce doming (depressing) of the upper ocean density surfaces inside AEs (CEs) (Gaube et al., 2015).

Moreover, the vertical potential density structures within SEs and SSEs are constructed using the Argo profiles, as shown in Fig 9. The result shows the distinct displacements of isopycnals between SEs and SSEs, which provides insight into the contrasting impacts on Chl-a distribution. The convex of isopycnals within SSAEs leads to the ascent of deeper water to the surface layer. This process facilitates the vertical transport of nutrients, promoting enhanced biological productivity and higher concentrations of Chl-a within SSAEs than SAEs. The vertical movement of water masses and the associated nutrient supply contribute to the favorable conditions for phytoplankton growth and the accumulation of Chl-a in SSAEs. Similarly, the concave of isopycnals within SSCEs leads to the subduction of surface water, resulting in lower Chl-a concentrations compared to SCEs.

By integrating these findings, we underscore the primary role of eddy–wind interaction in driving SSE formation, while the distinctive isopycnal displacements illuminate the underlying mechanisms contributing to varying Chl-a characteristics within these eddy types.

2. The manuscript concluded that SSEs account for nearly 50% of the total eddies, which needs further consideration. The sea surface temperature can be easily disturbed by environment, such as wind speed. Therefore, identifying SSEs according $SSTA<0$ (or $SSTA>0$) may increase the noises from low-energy eddies. It is suggested to set threshold for SSEs identification, such as amplitude, lifetime, which should increase the accuracy of SSEs identification.

**Response:** Thanks for the suggestion. Considering the resolution and precision of the SSHA product (Pujol et al., 2016), individual eddies with amplitudes $\geq$ 2 cm and radii $\geq$ 35 km are selected to avoid the noises from low-energy eddies in the study. Consequently, it is worth noting that the proportion of SSEs declined from the initial 44% to the current 39%. This adjustment is a direct outcome of utilizing the refined threshold dataset. Subsequent to this refinement, we have replotted Figures 4-9 and made appropriate modifications to the numerical values within Table 1 based on the updated data.

3. The paper proposes an identification method for SEs and SSEs using deep learning, along with validation and analysis of their temperature and chlorophyll characteristics. Consider refining the title to align more accurately with the manuscript's content.

**Response:** Thanks for the suggestion. The title has been revised to "Impact of Surface and Subsurface-Intensified Eddies on Sea Surface Temperature and Chlorophyll-a in the Northern Indian Ocean Utilizing Deep Learning."

4. Line 104: Reword 'as described by Assassi et al. (2016)' to 'as described in the study by Assassi et al. (2016)'.

**Response:** Revised as suggested.

5. Line 114-117: Has the sign of SS$\rho$/SSHA been previously used as an indicator to distinguish SEs and SSEs in any studies? Please provide references if available.

**Response:** Since SS$\rho$ cannot be directly measured from remote sensing observations. Instead, at first order, SS$\rho$ are primarily influenced by SST variations, which can be observed remotely. Therefore, the sign of SST/SSHA has been successfully used as an indicator to distinguish SEs and SSEs in previous studies (Greaser et al., 2020; Trott et al., 2019; Wang et al., 2019). Detailed information has been added in Section 2.2.1 of the revised manuscript.

6. Line 145: Please include the formula for the dice loss function.

**Response:** The formula for the dice loss function can be seen in the following:

$$Loss = 1 - Dicecoef(P, G) \tag{1}$$

The dice coefficient is a popular cost function for segmentation problems in deep learning. Given the predicted segmentation P and the ground truth region G, the dice coefficient is calculated as:

$$Dicecoef(P, G) = \frac{2|P \cap G|}{|P| + |G|} \tag{2}$$

where |.| is the sum of elements in the area. A good segmentation result is explained by a dice coefficient close to 1. A low dice coefficient (near 0) indicates poor segmentation performance. Detailed information has been added in Section 2.2.1 of the revised manuscript.

7. Line 146: What is the specific definition of accuracy for the DL-based model? Clarify this point.

**Response:** The categorical accuracy is used to estimate the eddy identification accuracy for the DL-based model. Categorical accuracy is a metric that calculates the mean accuracy rate across all predictions for multi-class classification problems. It is defined as follows:

$$Categorical\ accuracy = \frac{TP+TN}{TP+TN+FP+FN} \tag{3}$$

where TP, TN, FP, and FN represent the number of true positives, true negatives, false positives, and false negatives, respectively. Detailed information has been added in Section 2.2.1 of the revised manuscript.

8. Line 167: Provide detailed information on the inversed distance weighting interpolation method.

**Response:** Inverse distance weighting (IDW) is a deterministic method for multivariate interpolation with a known scattered set of points. The assigned values to unknown points are calculated with a weighted average of the values available at the known points. In the study, the temperature and potential density anomalies within 1.5R of mesoscale eddies were interpolated into 0.1R × 0.1R grid points up to a horizontal distance of 1.5R by the IDW interpolation method (Bartier & Keller, 1996) at each depth level (Dong et al., 2017; Sun et al., 2019; Yang et al., 2013). For each grid point, Argo profiles located within the horizontal range of 0.1R are set the weight value:

$$w_i = e^{-(\frac{d}{R})^2} \tag{4}$$

where d denotes the distance from the profile to the grid point. The final temperature or potential value at each grid point, $N_{grid}$, is calculated from the profile values $N_i$ as:

$$N_{grid} = \frac{\sum w_i N_i}{\sum w_i} \tag{5}$$

Detailed information has been added in Section 2.2.2 of the revised manuscript.

9. Line 258: Revise 'to accurately determine the most intense core's location' to 'to determine the location of the most intense core accurately.'

**Response:** Revised.

10. Figure 5: The Chl-a anomalies induced by SSAEs and SSCEs displayed in the current color bar are not easily discernible. It is recommended to modify the color bar to enhance the visibility of the differences.

**Response:** The color bar in Figure 5 has been revised as suggested.

**Reference**

Badin, G., Tandon, A., & Mahadevan, A. (2011). Lateral mixing in the pycnocline by baroclinic mixed layer eddies. *Journal of Physical Oceanography, 41*(11), 2080-2101.

Bartier, P. M., & Keller, C. P. (1996). Multivariate interpolation to incorporate thematic surface data using inverse distance weighting (IDW). *Computers & Geosciences, 22*(7), 795-799. https://doi.org/10.1016/0098-3004(96)00021-0

Dong, D., Brandt, P., Chang, P., Schütte, F., Yang, X., Yan, J., & Zeng, J. (2017). Mesoscale Eddies in the Northwestern Pacific Ocean: Three-Dimensional Eddy Structures and Heat/Salt Transports. *Journal of Geophysical Research: Oceans, 122*(12), 9795-9813. https://doi.org/10.1002/2017jc013303

Gaube, P., Chelton, D. B., Samelson, R. M., Schlax, M. G., & O'Neill, L. W. (2015). Satellite observations of mesoscale eddy-Induced Ekman pumping. *Journal of Physical Oceanography, 45*(1), 104–132. https://doi.org/10.1175/jpo-d-14-0032.1

Greaser, S. R., Subrahmanyam, B., Trott, C. B., & Roman-Stork, H. L. (2020). Interactions between mesoscale eddies and synoptic oscillations in the Bay of Bengal during the strong monsoon of 2019. *Journal of Geophysical Research: Oceans, 125*(10), e2020JC016772. https://doi.org/10.1029/2020JC016772

McGillicuddy, D. J., Jr. (2015). Formation of Intrathermocline Lenses by Eddy-Wind Interaction. *Journal of Physical Oceanography, 45*(2), 606-612. Article. https://doi.org/10.1175/jpo-d-14-0221.1

Meunier, T., Tenreiro, M., Pallàs-Sanz, E., Ochoa, J., Ruiz-Angulo, A., Portela, E., et al. (2018). Intrathermocline eddies embedded within an anticyclonic vortex ring. *Geophysical Research Letters, 45*(15), 7624-7633.

Pujol, M.-I., Faugère, Y., Taburet, G., Dupuy, S., Pelloquin, C., Ablain, M., & Picot, N. (2016). DUACS DT2014: the new multi-mission altimeter data set reprocessed over 20 years. *Ocean Science, 12*(5), 1067–1090. https://doi.org/10.5194/os-12-1067-2016

Sun, B., Liu, C., & Wang, F. (2019). Global meridional eddy heat transport inferred from Argo and altimetry observations. *Scientific Reports, 9*(1), 1345. https://doi.org/10.1038/s41598-018-38069-2

Thomas, L. N. (2008). Formation of intrathermocline eddies at ocean fronts by wind-driven destruction of potential vorticity. *Dynamics of Atmospheres and Oceans, 45*(3-4), 252-273.

Trott, C. B., Subrahmanyam, B., Chaigneau, A., & Roman-Stork, H. L. (2019). Eddy-induced temperature and salinity variability in the Arabian Sea. *Geophysical Research Letters*. https://doi.org/10.1029/2018GL081605

Wang, Z.-F., Sun, L., Li, Q.-Y., & Cheng, H. (2019). Two typical merging events of oceanic mesoscale anticyclonic eddies. *Ocean Science, 15*(6), 1545-1559. https://doi.org/10.5194/os-15-1545-2019

Yang, G., Wang, F., Li, Y., & Lin, P. (2013). Mesoscale eddies in the northwestern subtropical Pacific Ocean: Statistical characteristics and three-dimensional structures. *Journal of Geophysical*

*Research: Oceans, 118*(4), 1906-1925. https://doi.org/10.1002/jgrc.20164

---

## Author Comment (AC2)

**Comment:** After the revision, most of my questions got feedback. The manuscript is well-organized and suitable for publication. Other simple questions which are just my curious points: Do the authors have any idea how many chlorophyll profiles from BGC-Argo are associated with SE and SSE eddies? Do SE and SSE exhibit any differences in the vertical chlorophyll distributions?

**Response:** To answer this question, we obtained the Biogeochemical Argo (BGC-Argo) floats from https://dataselection.euro-argo.eu/. BGC-Argo floats are equipped with conductivity-temperature-depth (CTD) sensors to measure physical variables and bio-optical sensors to measure biogeochemical variables. For each BGC-Argo profile, we selected the highest-level data mode (delayed mode), produced later (over 1 year), and required control and validation by a scientific expert. Only profile data flagged as good quality were considered in the study. In addition, we conducted quality control on chlorophyll-*a* (Chl-*a*) profiles. First, a three-point moving median filter was applied on each profile to remove spikes (Bisson et al., 2019; Haëntjens et al., 2020). Next, we followed the calibration procedure of Roesler et al. (2017) and Haëntjens et al. (2020) to adjust the Chl-*a* data using the following equation:

$$\text{Chl-}a' = (\text{Chl-}a - 0.019) / 2.32$$

Where Chl-*a*' represents the Chl-*a* profiles used in the study and Chl-*a* represents the BGC-Argo Chl-*a* profiles that remove spikes. After the above data preprocessing, we obtained 399 (491) BGC-Argo profiles within 1.5 radii of AEs (CEs) in the North Indian Ocean during 2000-2015. The spatial distribution of these BGC-Argo profiles is shown in Figure 1.

[Figure]

**Figure 1.** The distribution of numbers of BGC-Argo profiles occurred in the 1°×1° bins within 1.5 radii of AEs (a) and CEs (b) in the North Indian Ocean during 2000-2015.

However, one should note that not all the above BGC-Argo profiles can be used to analyze Chl-*a* within eddies. Quality control was applied to eddy-collocated BGC-Argo floats using the following criteria: (1) Chl-*a* data from the upper 10 m were excluded from analyses because large variability and high uncertainty were observed there (Su et al., 2021); (2) Besides, each profile must contain at least one data point at a depth of 200 m or greater. It is because the Chl-*a* is generally located at the base of the euphotic layer (50 - 200 m) in the North Indian Ocean (Mignot et al., 2014); (3) There are more than 5 observations between 10 m and 200 m. As a result, only 30 (45) BGC-Argo profiles within 1.5R of AEs (CEs) meet the above criteria. Among them, 18 (12) BGC-Argo profiles were found within 1.5R of SAEs (SSAEs), while 32 (13) BGC-Argo profiles were found within 1.5R of SCEs (SSCEs).

Despite the small number of BGC-Argo profiles, we can see the differences in Chl-*a* profiles between SAEs and SSAEs or SCEs and SSCEs. As shown in Figure 2a, Chl-*a* induced by SSAEs is significantly greater than that caused by SAEs in the upper 30 m. Besides, Chl-*a* induced by SSCEs is significantly less than that caused by SCEs in the upper 50 m (Figure 2b). Such a result is consistent with the distinct displacements of isopycnals within SSAEs and SSCEs shown in Figure 9 in the manuscript. The convex of isopycnals within SSAEs leads to the ascent of deeper water to the surface layer. This process facilitates the vertical transport of nutrients, promoting enhanced biological productivity and higher concentrations of Chl-*a* within SSAEs than SAEs. The vertical movement of water masses and the associated nutrient supply contribute to the favorable conditions for phytoplankton growth and the accumulation of Chl-*a* in SSAEs. Similarly, the concave of isopycnals within SSCEs leads to the subduction of surface water, resulting in lower Chl-*a* concentrations compared to SCEs.

[Figure]

**Figure 2.** Mean (solid line) and standard deviation (shadow) values of Chl-*a* profiles for SAEs and SSAEs (a), and SCEs and SSCEs (b) in the North Indian Ocean.

According to the above results, we make the following modifications to the

manuscript: (1) add detailed information on BGC-Argo data, data preprocessing, and quality control in Section 2.1 Data; (2) add Figure 2 and its description and explanation in Section 4 Discussion; (3) add following references to the References part.

**References**

Bisson, K., Boss, E., Westberry, T., & Behrenfeld, M. (2019). Evaluating satellite estimates of particulate backscatter in the global open ocean using autonomous profiling floats. *Optics Express, 27*(21), 30191-30203. https://doi.org/10.1364/OE.27.030191

Haëntjens, N., Della Penna, A., Briggs, N., Karp-Boss, L., Gaube, P., Claustre, H., & Boss, E. (2020). Detecting Mesopelagic Organisms Using Biogeochemical-Argo Floats. *Geophysical Research Letters, 47*(6), e2019GL086088. https://doi.org/https://doi.org/10.1029/2019GL086088

Mignot, A., Claustre, H., Uitz, J., Poteau, A., d'Ortenzio, F., & Xing, X. (2014). Understanding the seasonal dynamics of phytoplankton biomass and the deep chlorophyll maximum in oligotrophic environments: A Bio-Argo float investigation. *Global Biogeochemical Cycles, 28*(8), 856-876. https://doi.org/10.1002/2013GB004781

Roesler, C., Uitz, J., Claustre, H., Boss, E., Xing, X., Organelli, E., et al. (2017). Recommendations for obtaining unbiased chlorophyll estimates from in situ chlorophyll fluorometers: A global analysis of WET Labs ECO sensors. *Limnology and Oceanography: Methods, 15*(6), 572-585. https://doi.org/https://doi.org/10.1002/lom3.10185

Su, J., Strutton, P. G., & Schallenberg, C. (2021). The subsurface biological structure of Southern Ocean eddies revealed by BGC-Argo floats. *Journal of marine systems, 220*, 103569. https://doi.org/10.1016/j.jmarsys.2021.103569